# Effect of Conformational Diversity on the Bioactivity of µ-Conotoxin PIIIA Disulfide Isomers

**DOI:** 10.3390/md17070390

**Published:** 2019-07-02

**Authors:** Ajay Abisheck Paul George, Pascal Heimer, Enrico Leipold, Thomas Schmitz, Desiree Kaufmann, Daniel Tietze, Stefan H. Heinemann, Diana Imhof

**Affiliations:** 1Pharmaceutical Biochemistry and Bioanalytics, Pharmaceutical Institute, University of Bonn, An der Immenburg 4, D-53121 Bonn, Germany; 2Department of Anesthesiology and Intensive Care, University of Lübeck, Ratzeburger Allee 160, D-23562 Lübeck, Germany; 3Eduard Zintl Institute of Inorganic and Physical Chemistry, Darmstadt University of Technology, Alarich-Weiss-Str. 4, D-64287 Darmstadt, Germany; 4Center for Molecular Biomedicine, Department of Biophysics, Friedrich Schiller University Jena and Jena University Hospital, Hans-Knöll-Str. 2, D-07745 Jena, Germany

**Keywords:** µ-conotoxin, PIIIA, voltage-gated sodium channel, disulfide connectivity, peptide folding, electrophysiology, molecular docking, molecular dynamics

## Abstract

Cyclic µ-conotoxin PIIIA, a potent blocker of skeletal muscle voltage-gated sodium channel Na_V_1.4, is a 22mer peptide stabilized by three disulfide bonds. Combining electrophysiological measurements with molecular docking and dynamic simulations based on NMR solution structures, we investigated the 15 possible 3-disulfide-bonded isomers of µ-PIIIA to relate their blocking activity at Na_V_1.4 to their disulfide connectivity. In addition, three µ-PIIIA mutants derived from the native disulfide isomer, in which one of the disulfide bonds was omitted (C4-16, C5-C21, C11-C22), were generated using a targeted protecting group strategy and tested using the aforementioned methods. The 3-disulfide-bonded isomers had a range of different conformational stabilities, with highly unstructured, flexible conformations with low or no channel-blocking activity, while more constrained molecules preserved 30% to 50% of the native isomer’s activity. This emphasizes the importance and direct link between correct fold and function. The elimination of one disulfide bond resulted in a significant loss of blocking activity at Na_V_1.4, highlighting the importance of the 3-disulfide-bonded architecture for µ-PIIIA. µ-PIIIA bioactivity is governed by a subtle interplay between an optimally folded structure resulting from a specific disulfide connectivity and the electrostatic potential of the conformational ensemble.

## 1. Introduction

Covalent linkage of cysteine residues by disulfide bonds is fundamental for the folding, stability, and function of many peptides and proteins [1,2,3,4]. The formation of correct disulfide linkages is particularly important for peptides and proteins with higher-order disulfide networks (>2 disulfide bonds) because incorrectly assembled disulfide bonds may directly interfere with the biological functions of these molecules.

The venom of marine cone snails is a rich source of small- to medium-sized disulfide-stabilized peptides, so-called conotoxins, most of which specifically target ion channels and receptors in the membranes of excitable cells [5]. Conotoxins are recognized for their pharmacological and therapeutic potential and they are models to study the impact of disulfide bonds on the structural stability of disulfide-rich peptides and proteins [4,6,7,8]. For example, it was shown that the biological activities and receptor specificities of some µ-, µO-, and α-conotoxins changed significantly if the native disulfide patterns of the peptides were experimentally modified [9,10,11,12].

The 3-disulfide-bonded µ-conotoxins specifically antagonize voltage-gated sodium channels (Na_V_ channels) making them attractive research tools as well as potential pharmacological lead structures [13]. µ-Conotoxins are so-called pore blockers because they bind to the extracellular pore vestibule and, hence, obstruct the permeation of sodium ions (Na^+^) through the channels [14]. While some µ-conotoxins, such as µ-GIIIA are very specific for skeletal muscle sodium channels (Na_V_1.4), µ-PIIIA—originally isolated from *Conus purpurascens*—also affects neuronal Na_V_ channels [10] and even some voltage-gated K^+^ channels [15]. Recently, we synthesized all 15 possible disulfide isomers (denoted in the current study as **1**–**15**) of the conotoxin µ-PIIIA (sequence: ZRLCCGFOKSCRSRQCKOHRCC-NH_2_), determined their structure and verified their disulfide bond patterns by applying a combination of MS/MS analysis and 2D NMR spectroscopy [16]. In the present study, these isomers and further analogs (Table 1) were analyzed in electrophysiological experiments for their potency to block the human skeletal muscle voltage-gated sodium channel Na_V_1.4. Interactions between the isomers and the channel protein were examined via molecular docking simulations. All the 21 PIIIA analogs (Table 1) were subjected to all-atom molecular dynamics (MD) simulations in solution while 3 selected isomer-channel complexes obtained from the docking simulations were subjected to further MD-based refinement for additional details of how the isomer-channel interaction stabilizes over time. The analysis of the 21 peptides (Table 1) revealed that the channel-blocking ability is determined by a complex interplay between conformational stability, orientations of the functionally significant side chains [17], and the electrostatic properties of the molecules. This study provides a conceptual framework to understand how different disulfide connectivities affect the structure of µ-PIIIA and ultimately its interaction with the pore of Na_V_1.4. 

## 2. Results

### 2.1. Bioactivity of 3- and 2-Disulfide-bonded µ-PIIIA Analogs at Na_v_1.4

Previously, we synthesized all 15 possible 3-disulfide-bonded isomers of the conopeptide µ-PIIIA (here termed **1**–**15**, Table 1) and determined their 3-dimensional structure [16]. Based on the native disulfide connectivity of isomer **2**, we now produced three additional 2-disulfide-bonded mutants of µ-PIIIA (termed **16**–**18**), each lacking one of the disulfide bonds of **2** (Table 1, Appendix A). Previous reports have shown that µ-PIIIA has the highest potency to interfere with skeletal muscle channel Na_V_1.4 although also being active on neuronal Na_V_ channels, such as Na_V_1.6 and Na_V_1.8 [18], and even on select voltage-gated K^+^ (K_V_) channels [15,19]. We therefore applied all 3-disulfide-bonded as well as the 2-disulfide-bonded isomers to HEK293 cells expressing human Na_V_1.4 channels and measured the impact of the toxin isomers on depolarization-elicited Na^+^ inward currents, expressed as the time constant for current block and the maximal block after long toxin exposure (Figure 1, Appendix A). Figure 1 summarizes the inhibitory activity of the most active isomers **1**, **2**, **7**, **11**, **12**, **14**, and **15**. As demonstrated in Figure 1A, currents mediated by human Na_V_1.4 were diminished by all isomers, albeit to different degrees. At a concentration of 10 µM, isomer **2**, which has the native disulfide connectivity of µ-PIIIA (C4-C16/C5-C21/C11-C22), was most effective in inhibiting Na_V_1.4 channels, followed by isomers **15**, **11**, **14**, **1**, **7**, and **12**. Analysis of the concentration dependence revealed that isomer **2** blocked human Na_V_1.4 with an apparent IC_50_ of 105.3 ± 29.9 nM, which is comparable to the value for the paralog rat Na_V_1.4 channels (103.2 ± 9.9 nM) [10]. As observed for some µ-conotoxins, even at a saturating concentration of 100 µM a small current component of 7.2 ± 2.2% remained, indicating that channel occupancy by isomer **2** does not eliminate Na^+^ conduction (Figure 1A,B). The remaining isomers (**15**, **11**, **14**, **1**, **7**, **12**) blocked Na_V_1.4 channels less potently than isomer **2**. In addition, the onset of block as estimated with single-exponential functions was substantially slower than for isomer **2**, thus precluding faithful assessment of channel block at lower concentrations than 10 µM and to determine the associated IC_50_ values (Figure 1C). Assessment of higher concentrations revealed that particularly isomers **15** and **7** are interesting: total current block at 10 and 100 µM was virtually identical (**15**: 48.2 ± 5.9% and 51.1 ± 2.2%, respectively; **7**: 32.3 ± 4.3% and 35.1 ± 4.6%, respectively), while the time constant characterizing the kinetics of onset of block, τ_block_, roughly scaled linearly with the concentration (**15**: 10 µM: 160.0 ± 34.9 ms; 100 µM: 4.7 ± 1 ms; **7**: 10 µM: 853.0 ± 195.0 ms; 100 µM: 59.0 ± 18.1 ms; Figure 1C). This result indicates saturated association of isomers **15** and **7** and the channel, with imperfect (about 50% and 35%, respectively) occlusion of the Na^+^ permeation pathway. In contrast, for isomers **11**, **14**, **1**, and **12** saturation of channel block was not apparent as increasing the concentration of the peptides from 10 µM to 100 µM also increased channel block (Figure 1C). The channel block of isomers with even lower potency (**3**–**6**, **8**, **10**, **13** and **16**–**18**) were analyzed at 10 µM. As shown in Appendix A, sample **13** was most active among this group, as it inhibited Na_V_1.4 by 24.6 ± 3.0%, followed by samples **4** (21.7 ± 3.5%), **9** (17.7 ± 1.7%), **3** (17.6 ± 3.6%), **17** (16.5 ± 0.5%), and **18** (16.5 ± 0.3%). Isomer samples **6**, **8**, **16**, **5,** and **10** diminished Na_V_1.4-mediated currents by less than 15% (**6**: 12.6 ± 1.3%, **8**: 11.4 ± 2.2%, **16**: 10.5 ± 1.3%, **5**: 6.7 ± 2.1%, **10**: 1.0 ± 3.0%), and thus were considered inactive under these conditions.

### 2.2. In Silico Toxin Binding Studies of the 3-Disulfide-Bonded µ-PIIIA Analogs at Nav1.4

Interactions of the µ-PIIIA isomers with the Na_V_1.4 channel were further investigated by docking isomers **1**, **2**, **4**, **7**, **11**–**15** (Table 1) to the channel (pdb ID 6AGF [20]) using the HADDOCK easy web interface (Figure 2, Appendix A). The remaining isomers **3**, **5**, **6**, **8**, **9**, **10**, and **16**–**18** were not further analyzed as they were considered to be inactive with respect to their poor pore blockage (Figure 1).

The docking results were clustered according to the interface-ligand root mean square deviations (RMSDs) and scored according to the HADDOCK scoring function (Appendix A). Multiple poses of the toxin isomers bound to the channel were observed and described in each case. The observations from the docking experiments were in line with the experimental observations indicating that the native and most active isomer **2** (Figure 1, Figure 2A) preferred a binding conformation that covered the central pore, while the moderately active ones chose conformations that left large portions of the pore exposed (isomers **11**, **12**, **14**, **1** and **7**) and/or did not insert as deeply into the pore (isomers **13**, **14** and **15**) as the native isomer **2** (Figure 1, Figure 2B,C, Appendix A). Isomer **2** was observed to reside closest to the selectivity filter residues located at the center of the pore in a similar position as recently found for the structurally related µ-conotoxin KIIIA occluding the pore with its central R14, while tightly binding to three out of the four channel subunits (Figure 2A). The moderately active isomers, were found to bind away from the center of the pore at the interface of the subunits I, III, and IV leaving only a single toxin residue close to the pore (except for isomers **15** and **4**, Figure 1, Figure 2B,C, Appendix A). Another general observation from our docking experiments was that the C- or N-terminus of the moderately active isomers was located closest to the selectivity filter region, again only allowing for an insufficient pore blockage by a single toxin residue.

However, at this point it shall be noted that the degree of pore blockage, which corresponds to the remaining current as revealed by the electrophysiological experiments, can only partly be rationalized from the docked pose of the isomers since the absolute degree of pore block could only be determined for isomers **2**, **15** and **7** (Figure 1 and Figure 2). Nevertheless, the binding poses of isomers **2**, **15** and **7** might explain their decreasing ability to block the pore (Figure 2) unveiling a smaller number of hydrogen bonds with residues close to the selectivity filter for **15** and **7** compared to the native isomer **2** (Figure 2). Moreover, **15** and **7** were found to only block the pore by their flexible *N*-terminal residues Z1 and R2 (Figure 2). In contrast to isomer **2** and **15**, the center of isomer **7** is not located above the pore leaving enough space for sodium ions to get close to the selectivity filter region, which might explain the even lower degree of pore block compared to **2** and **15** (Figure 2.).

### 2.3. MD- and Molecular Electrostatic Potential (MEP)-Based Analysis

#### 2.3.1. Analysis of 3-Disulfide-bonded µ-PIIIA Isomers **1**–**15**

With the biological activity of the 3-disulfide-bonded µ-PIIIA isomers **1**–**15** and the 2-disulfide-bonded analogs **16**–**18** determined via electrophysiological assessment, and their probable binding modes estimated by molecular docking, we conducted MD simulations to determine the impact of the distinct disulfide bonding on conformational stability and dynamics. From the electrophysiological investigation, it was evident that the native isomer **2** (C4-C16/C5-C21/C11-C22) of µ-PIIIA showed the highest activity by blocking >90% of channel current (Figure 1). The biological activity dropped to ~50% block for isomer **15** (C4-C22/C5-C21/C11-C16) and even below for all other active analogs **11**, **14**, **1**, **7**, and **12**. Based on this premise, two independent 400-ns MD simulations were conducted on each investigated disulfide isomer. The most obvious finding from the MD simulations was that isomer **2** was the most thermodynamically stable among the 3-disulfide-bonded analogs studied herein. All the structural and energetic properties computed from the two independent simulations (Appendix A) were almost identical for the trajectories of isomer **2**, while the other 14 analogs, despite converging to similar conformations and measured average properties, took diverging paths to equilibration. The ΔG_solv_ estimated from atomic hydrophobicity and surface solvent accessibility [21,22] was the lowest for **2** among all the 3-disulfide isomers (Appendix A). This is an indicator that the folded conformation of **2** is energetically slightly preferred over e.g., isomers **15**, **11**, and **14**. From Figure 3, Appendix A it is evident that the structure of isomer **2** exhibits good backbone stability with the average root mean square deviation (RMSD) value between the two trajectories of 2.55 ± 0.47 Å (Appendix A). From the structural superimpositions (Figure 3A) and root mean square fluctuation (RMSF) plot (Figure 3E, Appendix A) it is evident that the leading three *N*-terminal residues possess greater flexibility than the rest of the structure as this part of the sequence is not constrained by disulfide bonds.

Comparing the structure and dynamics of **7** with **2** reveals that in terms of backbone mobility, the key difference again lies in the flexibility of the *N*-terminal residues (Figure 3). From the superimposed structures in Figure 3B, we noticed that the C4-C11 disulfide bond renders its bracketed residues to adopt an anti-parallel β-sheet conformation (Figure 3B). This secondary structural adoption reorients the R2 residue towards the core of the peptide where it forms intramolecular H-bonds with C11 at the backbone in 40% of the trajectory, thus keeping it away from possible contacts with the channel. Added to this, **7** has the largest Molecular Electrostatic Potential (MEP) among all the studied isomers at 11.49 kJ/mol keeping the peptide bound to the channel by electrostatic attraction. The reduced channel-blocking ability of **7** primarily results from the orientations of its basic side chain residues, which are significantly different from the native **2** (Figure 2F). Moreover, the key residue R12 was involved in intramolecular H-bonds with Q15 for ~25% of the simulation taking it out of play for making contacts with the channel. **10** (C4-C16/C5-C22/C11-C21), whose disulfide architecture is the closest to that of **2** (Figure 2C), is surprisingly the isomer with the lowest measured bioactivity (Suppl. Figure 1). The isomer’s backbone undergoes an average change of 3.3 ± 0.6 Å RMSD to reach an equilibrated state (Figure 3D) with the average fluctuations of all its residues being much higher than those of **2** (Figure 3E). Structural superimpositions of **10** on **2** reveals clear misalignments between their key arginine and lysine residues. Furthermore, a series of intramolecular H-bonds by these key residues R2, R12, R14 and K17 for up to ~60% of total simulation time render them unavailable for interacting with the channel residues. In addition, **10** has the lowest MEP value at 6.76 kJ/mol leaving it at a disadvantage in terms of favorable electrostatics to retain it near the channel pore. Despite the stable backbone conformations observed for **8**, it still falls in the inactive set for the same reason as **10**, the formation of intramolecular H-bonds between the side chains of key residues that take them out of play from channel binding. In **8**, H-bond interactions between R12 and Z1 for ~65% of the simulation, K9 and Q15 for ~8% of the simulation, R20 and K17 for ~10% of the simulation, K17 and C21 for ~24% of the simulation and the side chain of R2 to its own main chain for ~5% of the simulation ensure that these residues are almost never available to make contacts with the channel. Structures from the MD trajectories were clustered based on a backbone RMSD threshold of 1 Å using the “single-linkage” algorithm to facilitate the selection of representatives out of isomers that sample diverse conformations. From Appendix A it is evident that some of the less active isomers (e.g., **3**, **4**, **5**, and **9**) result in many clusters and their largest clusters represent only a small percentage of the trajectory. The corresponding RMSD plots (Appendix A) also convey the same message. It can be deduced that the extreme conformational instability in isomers **3** (C4-C5/C11-C16/C21-C22), **4** (C4-C5/C11-C21/C16-C22), **5** (C4-C5/C11-C22/C16-C21), **6** (C4-C11/C5-C16/C21-C22), and **9** (C4-C16/C5-C11/C21-C22), arise from their disulfide bonding architecture of pairing between adjacent cysteines with no crosslinking disulfides to stabilize the segment. Hence they all fall into the inactive category. Visual inspection of the MD trajectory of these isomers revealed excessive rotational and translational motion which can be quantified by their RMSDs averaging at ~ 6 Å from their initial conformations (Appendix A). In terms of channel binding this means that these isomers are never found in a conformation to form stable contacts with the channel residues and leave the pore area constantly exposed. For the isomers in the active category (**1**, **7**, **11**, **12**, **14** and **15**), a combination of moderate to good backbone stability and favorable orientations of at least some of the key residues result in conformations that could partially but stably block the channel pore.

Additionally, 400-ns long refinement MD simulations of the isomer-channel complexes were conducted with a membrane system embedding the channel for the isomers **2**, **7**, and 15. All of the simulation parameters between the these isomer-channel complex MD simulations and the isomer in solution simulations were kept identical to maintain uniformity of the systems and comparability of results. All the 3 isomer-channel systems had attained equilibration providing molecular level insights into the toxin-channel interactions and how this related to the extent of pore block obtained from the electrophysiological studies. Differences in the dynamic behavior of these three isomers between their unbound and channel-bound forms were measured in terms of average RMSD, radius of gyration (Rg), and solvent accessible surface area (SASA) values (Appendix A). As expected, the mean SASA values of the channel-bound forms of the isomer were comparatively lower than their unbound counterparts (Appendix A) owing to the fact that some part of the peptide’s solvent accessible surface is not available to the solvent but to the channel to interact with. An estimate of the binding energy (E_bind_) was computed from the simulation and the values (Appendix A) showed a clear correlation with the binding potency quantified by the electrophysiological results. Isomer **2**, the best binder among all the isomers had an E_bind_ of -99.56 kJ/mol, while isomer **15** had -144.71 kJ/mol and finally isomer **7** with an E_bind_ of -332.83 kJ/mol (Appendix A), preserving the trend observed from the electrophysiological experiments. As observed in the docking simulations, isomer **2** was found to occupy the central region of the channel with its R14 residue occluding the pore facilitating channel block at the beginning of the simulation. Over the course of the 400-ns simulation, it was observed that the isomer underwent a conformational change accounting to an average backbone RMSD of 3.2 Å. ~95% of conformations sampled by the isomer had the R14 residue placed directly above the channel pore persistently forming H-bonding interactions with the channel residue E184 making sure that the pore was covered. Additional isomer-channel H-bonding interactions namely Z1-D472, R12-L475, and R12-N474, stabilized the residence of the isomer **2** over the channel pore. For ~5% of the simulation, the isomer adopted a conformation where the sidechain of R14 reoriented away from its original position covering the pore and formed intramolecular H-bonds with the S13 residue on its own backbone. During this event it was observed that the channel pore was clearly exposed in a way that allows the passage of ions. This could be a reason as to why the channel was not blocked to 100% by this isomer (Figure 1). The docked conformation of isomer **15** had its *N*-terminal oriented towards the interior with Z1 and R2 residues occupying the area over the pore. A H-bond between the residues R2 of the isomer and D500 of the channel added to the initial stability of the configuration. As the simulation progressed, the isomer adopted an altered conformation in which the channel pore was only partly covered, and no single residue occluded the pore. The isomer underwent an average conformational change of 4.36 Å backbone RMSD during the simulation. For isomer **7**, the starting structure (docked complex conformation) in simulation had the isomer oriented over the channel pore with the Z1 residue found occluding the pore forming a H-bond with the channel residue G773. This conformation was different from what was observed in the MD simulation of this isomer in solution wherein the *N*-terminal residues formed an anti-parallel ß-sheet (Figure 3B). During the simulation it was observed that isomer 7 quickly (under 5 ns) adopted a similar conformation to that in solution forming anti-parallel ß-sheets between residues C5-O8 and K9-C11, pulling the Z1 residue away from the channel pore. Hence the channel pore remained exposed for large parts of the simulation. A small fraction of the conformations (~15%) sampled by the isomer involved the R2 residue stretching out over the pore providing the partial block. For the rest of the simulation this residue was found H-bonded to T772 which is a part of the loop structure of segment II of the channel, leaving the pore constantly exposed. The isomer had an overall RMSD of 1.69 Å during the simulation indicating that it was stable in its bound state although it did not block the pore sufficiently. It was also observed that this isomer’s most preferred conformation, where the pore remains exposed, is different by 5.13 Å from its less preferred conformation where pore block was observed. This could further relate to the large time constant of block observed for this isomer because a large conformational change was required for channel block to be initiated. Finally, the per-residue contribution to the solvent accessible surface area (SASA_res_) was computed for the three isomers and compared against their unbound forms (Appendix A). Residues found close to the channel pore (e.g., R14 for isomer **2**, Z1 and R2 for isomer **15** and Z1 for isomer **7**) had lower SASA_res_ values compared to their unbound versions, serving as a passive indicator to their involvement in and the extent of pore blockage (Appendix A). The observations from these simulations serve as a guide to understanding the molecular details of how the difference in disulfide connectivity between these three selected isomers influence the way they bind to the channel, thereby defining the function.

#### 2.3.2. Analysis of 2-Disulfide-bonded µ-PIIIA Isomers **16**–**18**

We studied the structural and functional contribution of the individual disulfide bonds to the native isomer **2** of µ-PIIIA. From the electrophysiological experiments, significant loss of channel-blocking activity among all the three 2-disulfide analogs (**16**, **17**, and **18**) was demonstrated. MD analysis revealed that all the three 2-disulfide analogs underwent a reasonable amount of backbone conformational change (Figure 4). Among the three, the backbone of isomer **18** lacking the C5-C21 disulfide bond evolved to be the most stable as observed from the RMSD profiles in Figure 4. **18** also had the least increase of per-residue fluctuations as shown in the RMSF comparison plots in Figure 4. This results in the marginally higher activity seen for **18** over **16** in Appendix A. In all the three 2-disulfide-bonded analogs, clear misalignment of the key basic residues was observed in comparison to the native 3-disulfide-bonded isomer **2** and hence the loss of bioactivity (Figure 4J, K, L). A recent report using a modified version of µ-PIIIA produced similar results with respect to assessing the role of individual disulfide bonds on the structure and bioactivity of µ-PIIIA [23], although in this study the *N*-terminal pyroglutamate residue was replaced by proline and respective cysteines were substituted by alanine residues.

#### 2.3.3. Analysis of 2-Disulfide-bonded µ-PIIIA Isomers Δ(C5-C21)2, Δ(C11-C21)4, and Δ(C5-C22)10

The concept of disulfide bonds holding a “structurally frustrated” peptide in its bioactive form is known from the literature [24]. A means to measure this inherent frustration induced by the disulfide bonding pattern can be studied by observing the dynamics of partially folded states of the peptide. Previous studies have also shown that this observation could give hints into the folding pathways that the peptide prefers to take as it folds [8]. Isomer **2**, **4**, and **10** have been selected as representatives of structurally well-defined (**2**), moderately flexible (**10**), and highly flexible (**4**) classes among the 15 µ-PIIIA isomers [16]. We created partially folded analogs of these isomers by arbitrarily removing the second disulfide bond in these structures creating the Δ(C5-C21)**2**, Δ(C11-C21)**4,** and Δ(C5-C22)**10** to observe the refolding preference of these partially folded states (Figure 5). Isomer Δ(C5-C21)**2** did not prefer to refold maintaining an average distance of 15 Å between the Sγ atoms of the opened C5-C21 disulfide bond, while in analog Δ(C5-C22)**10** at several time points the Sγ atoms of the opened C5-C22 disulfide bond came up to 5 Å close but eventually sampled unfolded conformations towards the end of the simulation (Figure 5). The analog Δ(C11-C21)**4** displayed an equal mix of both unfolded and folded states with respect to its parent isomer **4** structure with the distances between the unbound Sγ atoms of the opened C11-C21 disulfide bond fluctuating heavily between 3 Å to 30 Å (Figure 5). This analysis confirms that the folding preferences between different disulfide isomers of the same sequence can be highly diverse, further supporting that µ-PIIIA does require all three of its disulfide bonds to remain structurally stable and bioactive.

## 3. Discussion

The significance of disulfide bonds in the stability and folding of peptides and proteins is well-established and accepted [25]. Beyond their involvement in proteins for thermodynamic and kinetic stability [26], disulfide bonds also play multifaceted roles in peptides with therapeutic potential [27]. While the so-called “native fold” is often what is studied with much rigor, we have in recent studies ventured into exploring all alternative folds using the 3-disulfide-bonded µ-PIIIA as target [17]. In the present study, we have clarified using a combination of electrophysiological and computational approaches that despite the native fold (isomer **2**) still carrying the highest bioactivity, 6 of the remaining 14 isomers are still bioactive in a range between 30%–50% of maximum current block compared to **2**. Due to its globular conformation, homogenous distribution of functionally significant basic arginine and lysine residues and their stable orientations [17,28,29] indicated by the lowest RMSF values among the 15 isomers studied, isomer **2** is both structurally and energetically the most favorable to optimally block the channel (Figure 3). It is also the largest populated isomer and the most easily formed with ΔG_solv_ computation [23] serving as an indicator of stability.

Our current findings help to clarify questions about ambiguities regarding the potencies of the different isomers. In an earlier study (Tietze et al., 2012) [10], isomer **3** was active (~50% as active as the native isomer), while it was inactive in the present study. However, taken into account the recent study by Heimer et al., 2018 [16], we are now able to state that **3**, when obtained from the oxidative self-folding approach, obviously contained other bioactive isomers, such as **2** or **15,** as confirmed by corresponding HPLC elution profiles of the individual mixtures [16]. In addition, these impurities explain why NMR analysis could not be obtained for this isomer in our first study (Tietze et al., 2012) [10] but could be elucidated upon targeted synthesis and purification of isomer **3** by Heimer et al. [16]. In the same study by Tietze et al., 2012 [10], isomer **1** was more active than the natively folded isomer **2**. Considering our recent report [16], this discrepancy can also be explained by the HPLC analysis of the isomer mixtures. In this recent report, it was demonstrated that isomer **1** cannot unambiguously be separated from **2** [16], which was also confirmed by impurities (unassigned) found in the NMR spectra, although the analysis of the structure of isomer **1** was still feasible at the time.

The negligible to no activity of **8** and **10** represents an interesting case of how structural stability does not always relate to bioactivity and adds evidence to the notion that folding proceeds in the direction that preserves function [29], and that even slight alterations of the disulfide architecture can at times be functionally inefficient. The current study also provides experimental proof for an earlier in silico study which suggested that native µ-PIIIA does certainly need all its 3 disulfide bonds to remain functional [8] compared to other conotoxins such as µ-KIIIA [7] and α-conotoxin Vc1.1 [30]. In summary, using electrophysiology, molecular docking, MD simulations, and MEP analysis, we have established a molecular, structural and electrostatic basis for the impact of disulfide connectivity on the bioactivity of µ-PIIIA isomers. We have additionally used simulations to explain the loss of bioactivity in the 2-disulfide-bonded µ-PIIIA analogs.

## 4. Materials and Methods

### 4.1. Peptide Synthesis and Purification

Synthesis, purification, and chemical and structural analysis of the 15 3-disulfide-bonded PIIIA isomers **1**–**15** (sequence: ZRLCCGFOKSCRSRQCKOHRCC-NH_2_, Appendix A) has been described previously [16]. Freeze-dried aliquots of known concentrations for each isomer, as determined by amino acid analysis and HPLC, were submitted to electrophysiological experiments. In addition, three 2-disulfide-bonded isomers **16**–**18** were prepared and subjected to biological activity testing as well. In brief, these µ-PIIIA-mutants were automatically assembled using 9-fluorenyl-methyloxycarbonyl (Fmoc) chemistry and couplings employing 2-(1H-benzotriazol-1-yl)-1,1,3,3-tetramethyluronium hexafluorophosphate (HBTU) and *N*-methylmorpholine/DMF (1:1) on a Rink-amide MBHA resin (0.53 mmol/g) and an EPS221 peptide synthesizer (Intavis Bioanalytical Instruments AG, Cologne, Germany). The different cysteine pairs were protected with Trt- and Acm-groups, or exchanged with Ser according to the desired disulfide connectivity (Appendix A). Peptide cleavage and concomitant side chain deprotection was accomplished with reagent K (phenol/thioanisole/ethanedithiol 1:1:0.5, 150 µl/100 mg resin) and TFA (95%, 1 mL/100 mg resin) for 3 h. The crude peptides were precipitated with diethyl ether, centrifuged, and washed several times with diethyl ether. Linear precursor peptides were purified as previously described [16] and chemically characterized (Appendix A). The oxidation of the 2-disulfide bonded peptides **16**–**18** was performed in a mixture of isopropanol/water/1 M HCl (31:62.5:7.5) with a final peptide concentration of 0.05 mM. Then, 1.1 equiv. iodine (0.1 M in MeOH) was added to the peptide solutions, and the reaction was stirred for 1 h at room temperature. After the first disulfide bond was closed, further 13.9 equiv. iodine (0.1 M in MeOH) was added to deprotect the Acm-group and to form the second disulfide bond. After the reaction was stirred for up to 1 h, the oxidation was stopped by the addition of an excess of ascorbic acid (1 M in water). Reaction progress was monitored via RP-HPLC and LC-ESI-MS. The products were freeze-dried and stored at −20 °C. The purification of the crude peptides was performed by semi-preparative RP-HPLC as reported [16]. Peptide purity and constitution was confirmed by analytical RP-HPLC, MALDI-TOF mass spectrometry and amino acid analysis (Appendix A) using methods and instruments as described previously [16]. Amino acid analysis was used for the determination of peptide concentrations in solution prior to the further experiments to ensure equal concentrations in analysis and biological testing. In general, µ-PIIIA derivatives were dissolved in double distilled water at a stock concentration of 10 µM and further diluted as described in the electrophysiological experiments.

### 4.2. Electrophysiological Experiments

Human SCN4A (encoding the Na_V_1.4 channel α subunit, UniProt ID P35499) on a plasmid with *Cytomegalovirus* (CMV) promoter was transiently expressed in HEK 293 cells as shown previously [10]. Co-transfection of a plasmid encoding the CD8 antigen ensured visual detection of transfected cells with CD8-specific Dynabeads (Deutsche Dynal, Hamburg, Germany). Currents were measured with the whole-cell patch clamp method 24–48 h after transfection [10]. The patch pipettes contained (in mM): 35 NaCl, 105 CsF, 10 EGTA (ethylene glycol bis(2-amino-ethylether)tetraacetic acid), 10 HEPES (pH 7.4 with CsOH). The bath solution contained (in mM): 150 NaCl, 2 KCl, 1.5 CaCl_2_, 1 MgCl_2_, 10 HEPES (pH 7.4 with NaOH). Holding potential was –120 mV, Na^+^ currents were elicited with depolarizing steps to −20 mV. Series resistance was corrected electronically up to 80%. Peptides, diluted in the bath solution, were applied focally to cells under consideration with a fine-tipped glass capillary. Time course of peak current decrease after peptide application was described with single-exponential functions.

### 4.3. Molecular Modeling and Docking Simulations

The toxin-channel binding was predicted by docking the lowest energy conformer of the NMR structures of the μ-PIIIA isomers **1**, **2**, **4**, **7**, **11**-**15** [16] to the recently solved structure of the human Na_V_1.4 voltage-gated sodium channel (pdb ID 6AGF [20]) using the Easy Interface of the HADDOCK online platform [31,32] (https://haddock.science.uu.nl/services/HADDOCK2.2/haddockserver-easy.html), a web service known to be suitable for handling more complex peptide ligand structures [32]. The receptor structure was energy-minimized in explicit solvent (TIP3 water) before the docking runs.

For the docking process, all toxin residues and the channel residues that are part of the channel’s upper surface were defined as “active”, because they were assumed to be able to form contacts with the toxin. As “passive” channel residues we defined all residues within a radius of 6.5 Å from the “active” residues. From the docking results, the best scoring structure from the highest scoring complex cluster was selected for further analysis (Appendix A). As the HADDOCK web interface cannot handle γ-pyroglutamic acid, glutamic acid was used instead, and was re-converted to γ-pyroglutamic acid, followed by a subsequent energy minimization step for further analysis.

Analysis and energy minimizations were performed using the YASARA molecular modeling software (YASARA structure, Vers. 18.3.23, YASARA Biosciences GmbH, Vienna, Austria) [33,34].

The energy minimizations of the toxin-channel complex were performed in explicit water (TIP3P) and the Particle Mesh Ewald (PME) method [35] in order to describe long-range electrostatics at a cut-off distance of 8 Å in physiological conditions (0.9% NaCl, pH 7.4), at a constant temperature (298 K) using a Berendsen thermostat, and with constant pressure (1 bar). The charged amino acids were assigned according to the predicted pKa of the amino acid side chains from the Ewald summation, and were neutralized by adding counter ions (NaCl). The YASARA2 force field was used for energy minimization by simulated annealing, including the optimization of the hydrogen bond network and the equilibration of the water shell, until system convergence was achieved.

The molecular graphics were created using YASARA (YASARA structure, Vers. 18.3.23, YASARA Biosciences GmbH, Vienna, Austria, www.yasara.org) and POVRay (Persistence of Vision Raytracer Pty. Ltd., Williamstown, Australia www.povray.org).

### 4.4. MD Simulations of the µ-PIIIA Isomers and Analogs

All-atom unbiased MD simulations of the 15 3-disulfide-bonded isomers (**1**–**15**), three 2-disulfide bonded analogs (**16**–**18**) and three partially folded 2-disulfide bonded models **2-2S**, **4-2S** and **10-2S** were conducted with the GROMACS 2018 program [36,37]. The 3D coordinates for isomers **1**–**15** were obtained from the NMR structural ensembles reported [16]. The first out of the 20 structures of the ensemble was taken as the starting conformation for MD simulations. Analogs **16**–**18** were modeled in YASARA [34] by replacing the respective cysteine residues by serine. These models were energy-minimized using the AMBER99SB-ILDN [38] force field (Amber 11 in YASARA) before being used for simulations. The partially folded conformations were created in GROMACS via the *pdb2gmx* module as earlier described [8]. All steps and parameters involved in preparing the MD system were taken from the previous work [8] that involved the simulation of the native isomer of µ-PIIIA, except that the simulation run times increased herein. Solvent equilibration for 50 ns in constant temperature and constant-volume NVT ensemble, and 50 ns in the constant temperature and constant-pressure NPT ensemble conditions were carried out with position restraints on the energy-minimized peptide prior to the production MD run. Production runs at 300 K were conducted for 400 ns with a 2 fs time step. Two independent simulations were conducted per peptide using the random seed method that assigns different initial velocities (from the Boltzmann distribution) to the system thereby ensuring adequate sampling of the conformational space. 10,000 frames from each trajectory were used for analysis. The RMSD of the protein backbone atoms, per-residue RMSF of all peptide atoms, the radius of gyration (Rg) of all peptide atoms, and the SASA were computed using analysis tools within GROMACS. The double cubic lattice method [39] with the default probe radius of 1.4 Å was used to compute the SASA as well as the per-residue contribution to the solvent accessible surface (SASA_res_). An estimate of solvation free energies (ΔG_solv_) from the atomic solvation energies per exposed surface area as implemented by Eisenberg et al. [21] were computed and plotted as a function of simulation time. Single-linkage clustering analysis was done on each trajectory using the backbone RMSD cut-off of 1 Å as the metric for the clustering. The centroid of the largest cluster from each trajectory was used as a representative for further analysis. Visualizations of the MD trajectories, molecular graphics and movies from the simulations were created in Visual Molecular Dynamics (VMD) (version 1.9.3) [40]. Plots were created using the Grace program (version 5.1.25).

### 4.5. MD Simulations of the µ-PIIIA Isomer-Channel Complexes

Owing to the ease of setting up a membrane simulation, YASARA was used in conducting 400-ns long MD simulations of isomers **2**, **7**, and **15** bound to the recently resolved cryo-electron microscopy structure (PDB 6AGF [20]) of the human voltage-gated sodium channel Na_V_1.4. The 3 isomers were selected based on a consensus obtained from the electrophysiological experiments. The preparation of the membrane embedded system was achieved by providing the docked isomer-channel complex as input to the *md_runmembrane.mcr* macro in YASARA. The program first identified the transmembrane helices by scanning the channel structure for secondary structure elements with hydrophobic surface residues to orient and embed a membrane composed of 1-palmitoyl-2-oleoyl-sn-glycero-3-phosphocholine (POPC) fatty acid residues. The protein was oriented such that the axis through the transmembrane helices is perpendicular to the membrane. A cubic simulation cell large enough to enclose the entire membrane was then drawn. The cell was extended by 15 Å on either side of the membrane so that the membrane was 30 Å larger than the protein ensuring that during simulation, the protein never sees its periodic image. The simulation cell was filled with the water molecules including enough number of Na^+^ and Cl^−^ ions to achieve a physiological concentration of 0.9% as a mass fraction. Next, successive steps of steepest descents energy minimizations followed by simulated annealing minimizations were done to remove bumps between the lipid molecules by deleting those lipids resulting in unfavorable geometries. The minimization steps were done iteratively until the potential energy of the system was optimized. Any solvent molecule found within the lipids was removed and the membrane was slowly packed closer by iteratively reducing the size of the simulation cell. As the final step, the membrane was artificially stabilized by position restraints and a pre-production simulation for 250 ps was done to equilibrate the solvent around the membrane and protein. The equilibrated system was composed of a total of ~163000 atoms including ~300 POPC molecules making up the membrane, within a cubic simulation cell with periodic boundaries of side ~120 Å was ready for the production MD simulation. A 2 fs time step was used to run the 400-ns long production MD simulations. Temperature was maintained at 300 K by a velocity rescaling algorithm and the pressure maintained at 1 bar by rescaling the volume. The cut-off for non-bonded forces was set at 8 Å. Analysis of the resulting trajectories were done via the standard macros within YASARA (*md_analzye.mcr* and *md_analyzeres.mcr*). An estimate of the binding energy E_bind_ was obtained using the Poisson-Boltzmann method [41,42] via the *md_analyzebindenergy.mcr* macro in YASARA by analyzing each frame in the 400-ns MD simulation. The E_bind_ was given by *E_bind_ = E_pot_Recept + E_solv_Recept + E_pot_Ligand+ E_solv_Ligand − E_pot_Complex − E_solv_Complex* where the subscripted “pot” and “solv” represent potential energy and solvation energy, respectively. Positive values from this result indicate better association between the ligand and receptor although negative values do not indicate no binding. The values obtained from the 3 simulations were compared against each other to estimate and compare the nature of the toxin-channel interactions for the 3 isomers (**2**, **7**, and **15**) in this study.

### 4.6. MEP Calculations

MEP calculations were made for multiple conformations for all peptides used in this study to help quantify the differences in the surface electrostatic properties. MEP calculations were done using the Poisson-Boltzmann method [41,42] via the Adaptive Poisson-Boltzmann Solver (ABPS) program [43] built into YASARA. The method uses an implicit solvation model wherein the solvent is treated as a high dielectric (ε = 80) continuum. To maintain consistency with the MD simulations, the AMBER99SB-ILDN force field was used. Frames from the last 300 ns of the MD trajectory were used as inputs for the MEP calculation.

## Figures and Tables

**Figure 1 marinedrugs-17-00390-f001:**
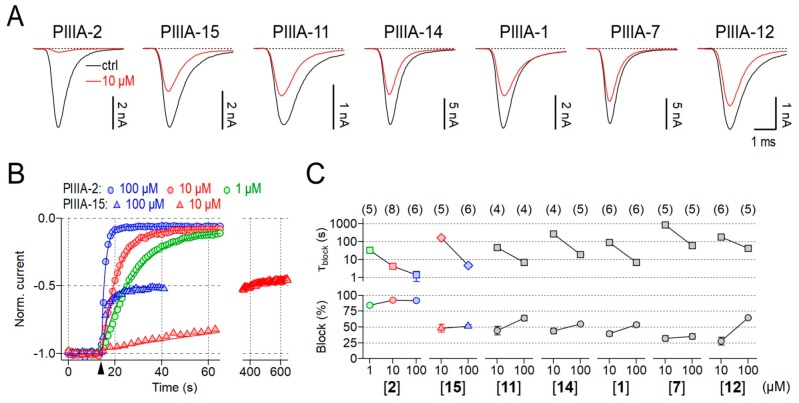
Disulfide isomers of µ-PIIIA differentially inhibit Na_V_1.4-mediated currents. (**A**) Representative current traces of transiently expressed Na_V_1.4 channels evoked at a test potential of −20 mV before (black, ctrl) and after (red) application of 10 µM of the indicated µ-PIIIA isomers. (**B**) Normalized peak current amplitudes obtained from repetitively evoked current responses were plotted as a function of time to follow the time course of current block mediated by various concentrations of µ-PIIIA isomers **2** (circles) and **15** (triangles). Continuous lines are single-exponential data fits used to characterize the onset of channel block. The arrowhead marks the start of peptide application. The time axis was split to illustrate that channel block by isomer **15** saturates at about 50% suggesting that isomer **15** seals the channel pore only partially. In contrast, isomer 2-mediated channel inhibition saturated at about 95%. (**C**) Steady-state block estimated from single-exponential fits of the time course of current inhibition (bottom) as well as the associated single-exponential time constant (top), describing the onset of channel block, for the indicated isomers and concentrations. Lines connect data points for clarity. Symbols and color-coding of data obtained with isomers **2** and **15** are as in (**B**). Numbers of individual experiments (n) are provided in parentheses.

**Figure 2 marinedrugs-17-00390-f002:**
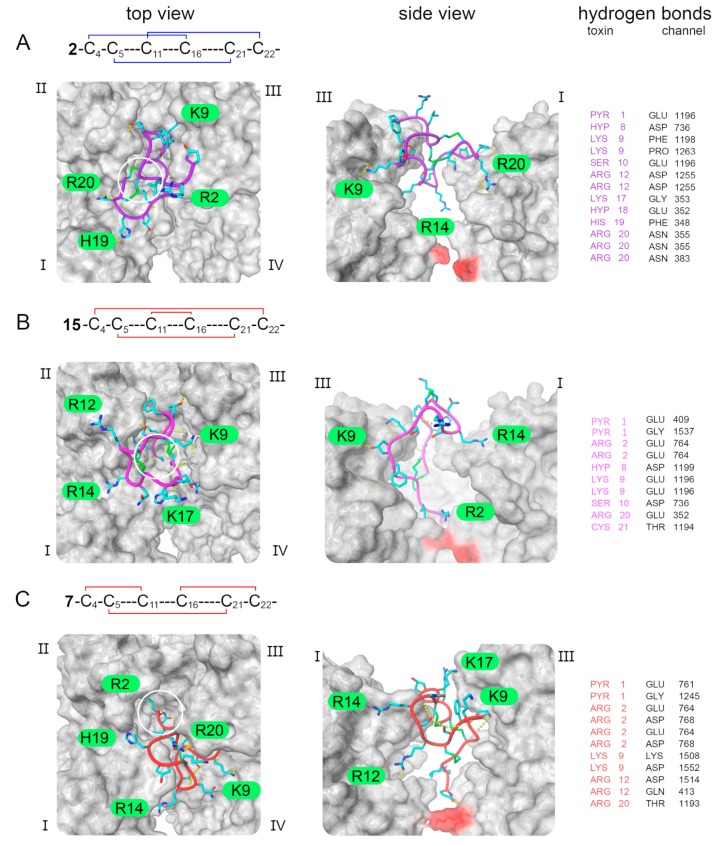
Visualization of µ-PIIIA–Na_V_1.4 complex conformations obtained from docking experiments for (**A**) the native isomer **2**, (**B**) isomer **15**, and (**C**) isomer **7**. Left panel – top view of the toxin-channel complex. Middle panel – side view of the toxin-channel complex. The four Na_V_1.4 domains are indicated and hydrogen bonds between the toxin and the channel are given as yellow dashed lines, specified in the right panel. The Na_V_1.4 channel surface (molecular surface) is given in gray, the selectivity filter motif (DEKA) is highlighted in red. The toxin is shown with side chain atoms present (coloring scheme: carbon – cyan, nitrogen – blue, oxygen – red, sulfur – green, backbone – isomer 2: purple, 15: pink, 7: red).

**Figure 3 marinedrugs-17-00390-f003:**
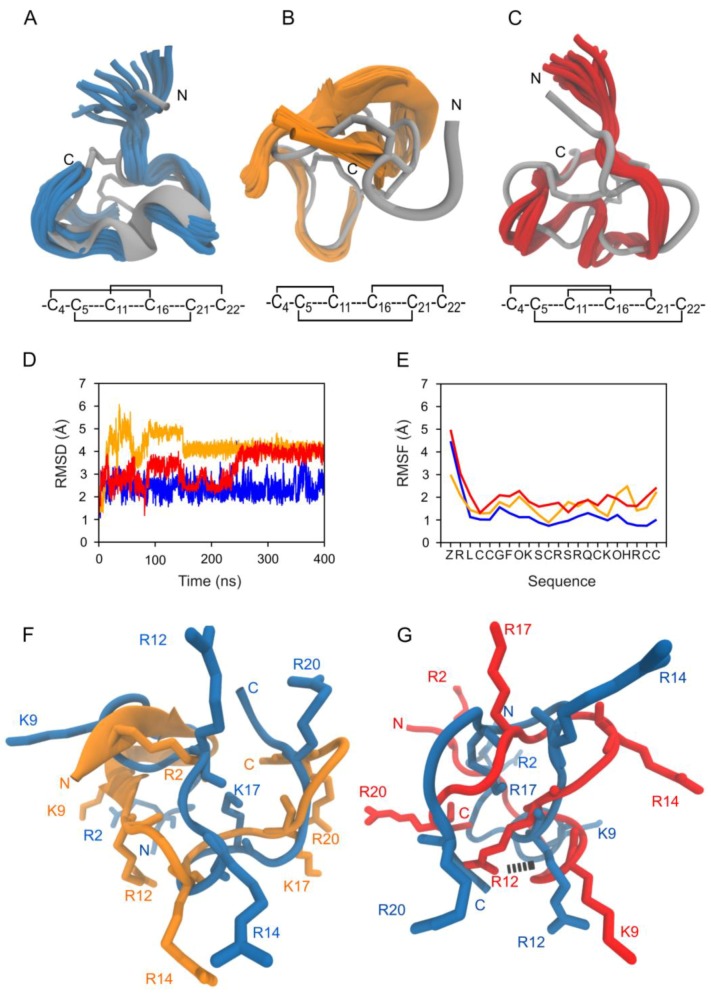
MD-based structural comparison of isomers **2**, **7**, and **10**. (**A**–**C**) 20 cartoon styled simulation snapshots of isomer **2** (blue), **7** (orange), and **10** (red) superimposed on their respective starting NMR structures (gray). The disulfide connectivity of each isomer is given below the individual isomer images. (**D**) Overlay of backbone RMSD progression plots for isomers **2** (blue), **7** (orange), and **10** (red) from a 400-ns MD simulation. (**E**) Overlay of per-residue root mean square fluctuation (RMSF) plots for isomers **2** (blue), **7** (orange), and **10** (red) from a 400-ns MD simulation. (**F**) Cartoon style structural superimposition of the representative structure of isomer **7** (orange) from MD simulation over the representative structure of isomer **2** (blue). The lysine and arginine residues significant for channel blocking are shown as sticks and labelled for both isomers. (**G**) Cartoon style structural superimposition of the representative structure of isomer **10** (orange) from MD simulation over the same of isomer **2** (blue). The lysine and arginine residues significant for channel blocking are shown as sticks and labelled for both isomers.

**Figure 4 marinedrugs-17-00390-f004:**
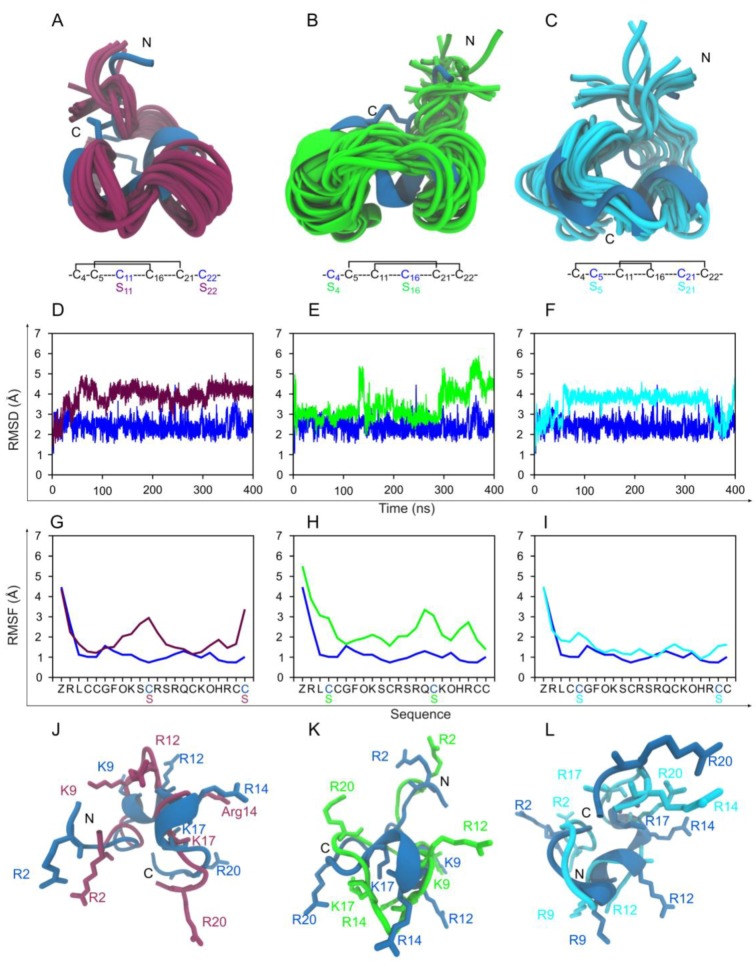
MD-based structural comparison of the 2-disulfide-bonded µ-PIIIA analogs **16**, **17**, and **18**. (**A**–**C**) 20 cartoon styled simulation snapshots of isomer **16** (maroon), 1**7** (green), and **18** (cyan) superimposed on the starting NMR structure of µ-PIIIA isomer **2** (blue). The disulfide connectivity and the C-S mutations of each isomer are given below the individual isomer images. (**D**) Overlay of backbone RMSD progression plots of isomer **16** (maroon) on the native isomer **2** (blue). (**E**) Overlay of backbone RMSD progression plots of isomers **17** (green) on the native isomer **2** (blue). (**F**) Overlay of backbone RMSD progression plots of isomers **18** (cyan) on the native isomer **2** (blue). (**G**) Overlay of per-residue RMSF plots of isomer **16** (maroon) on the native isomer **2** (blue). (**E**) Overlay of per-residue RMSF plots of isomer **17** (green) on the native isomer **2** (blue). (**F**) Overlay of per-residue RMSF plots of isomers **18** (cyan) on the native isomer **2** (blue). (**J**–**L**) Cartoon styled representative structures from MD for isomers **16** (maroon), **17** (green) and **18** (cyan) superimposed on the structure of the native isomer **2** (blue). The functionally significant lysine and arginine residues are shown as sticks and labelled.

**Figure 5 marinedrugs-17-00390-f005:**
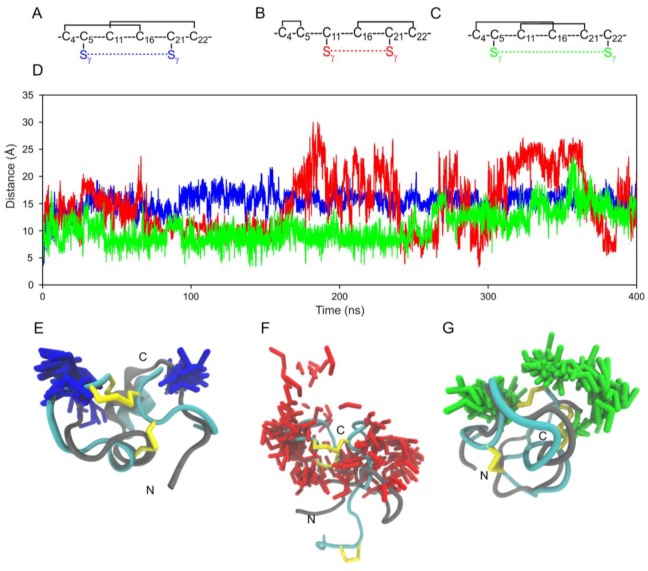
MD-based disulfide distance analysis of partially folded µ-PIIIA analogs Δ(C5-C21)**2**, Δ(C11-C21)**4** and Δ(C5-C22)**10**. (**A**–**C**) Disulfide connectivity (black solid lines) of the isomers. The colored dotted lines (blue – Δ(C5-C21)**2**, red – Δ(C11-C21)**4**, and green – Δ(C5-C22)**10**) represent the disulfide bonds opened *in silico*. (**D**) Overlay of distance plots between the *in silico* opened disulfide bonds (blue – Δ(C5-C21)**2**, red – Δ(C11-C21)**4**, and green – Δ(C5-C22)**10**). (**E**–**G**) The cyan cartoon structures represent the NMR structures of the isomers **2**, **4**, and **10** with their disulfide bonds shown as yellow sticks. The overlaid gray cartoon is a single MD snapshot of the partially folded 2-disulfide-bonded versions of the isomers with 100 conformations of the opened disulfide bond versions (blue – Δ(C5-C21)**2**, red – Δ(C11-C21)**4**, and green – Δ(C5-C22)**10**) to represent the distances sampled between them during simulation.

**Table 1 marinedrugs-17-00390-t001:** Nomenclature, disulfide connectivity, type of structure, and source references for the peptides used in this study.

Isomer Nomenclature	Number of Disulfides	Disulfide Connectivity	Type of Structure	Source Reference(s)
**1**	3	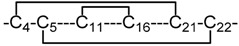	NMR	Heimer et al. [16], Tietze et al. [10].
**2**	3	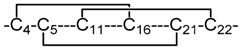	NMR	Heimer et al. [16], Tietze et al. [10].
**3**	3	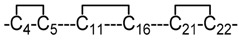	NMR	Heimer et al. [16], Tietze et al. [10].
**4**	3	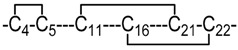	NMR	Heimer et al. [16].
**5**	3	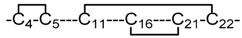	NMR	Heimer et al. [16].
**6**	3	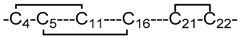	NMR	Heimer et al. [16].
**7**	3	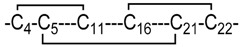	*In silico* model	Heimer et al. [16].
**8**	3	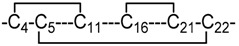	NMR	Heimer et al. [16].
**9**	3	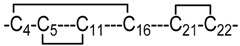	NMR	Heimer et al. [16].
**10**	3	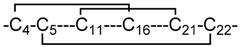	NMR	Heimer et al. [16].
**11**	3	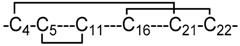	NMR	Heimer et al. [16].
**12**	3	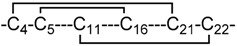	*In silico* model	Heimer et al. [16].
**13**	3	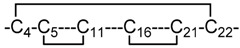	*In silico* model	Heimer et al. [16].
**14**	3	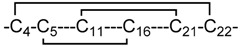	NMR	Heimer et al. [16].
**15**	3	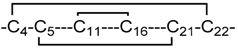	NMR	Heimer et al. [16].
**16***	2	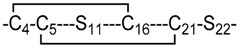	*In silico* model	Current study
**17***	2	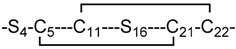	*In silico* model	Current study
**18***	2	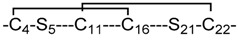	*In silico* model	Current study
Δ(C5-C21)**2**^#^	2	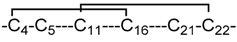	*In silico* model	Current study
Δ(C11-C21)**4**^#^	2	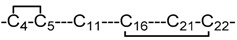	*In silico* model	Current study
Δ(C5-C22)**10**^#^	2	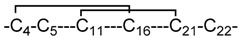	*In silico* model	Current study

* Mutated disulfide bonds replaced by serine residues. ^#^ Isomers with in silico disulfide bond opening yielding two SH-groups.

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
