# Peer review of "Effect of Conformational Diversity on the Bioactivity of µ-Conotoxin PIIIA Disulfide Isomers"

_marinedrugs, 2019, doi:10.3390/md17070390_

Round 1

Reviewer 1 Report

This manuscript employed solution NMR to characterise the structures of a range of isomers of the mu-conotoxin PIIIA with varying disulphide connectivities and electrophysiology experiments to determine their blockade of the sodium channel Nav1.4. It was determined that the native fold, with C4-16, C5-21 and C11-22 disulfide linkages, exhibited the highest bioactivity, although partial block was observed for several other non-native isomers.  Molecular docking and simulations were performed to determine the binding modes of native and disulphide isomers of PIIIA at Nav1.4 using a recently-solved structure of the channel, which provided molecular-scale rationalisations of the electrophysiology results.

The methodologies are sound, the results are well-supported by experiments and to some extent the computational data, and the manuscript is well-presented and clearly written. I recommend publication, with some minor revisions/additions:

-          Can the authors provide further details regarding the PIIIA-Nav1.4 complex simulations? It was stated that “All parameters…. were maintained identical between all simulated systems.” However, details regarding the system set-up, including the protocol used for channel embedding into the bilayer, equilibration, pressure coupling, number of lipids in each leaflet, and other standard methodological details for membrane simulations should be included.

-          The membrane system was described as consisting of “~ 100 Palmitoylethanolamide (PEA) residues”. Why was this single-chain fatty acid chosen rather than other, more common double-tailed lipids such as POPC, DOPC, etc? Additionally, 100 single-chain lipids seems like a relatively small number, leaving only roughly 50 lipids per leaflet to surround the channel as well as provide a buffer space between mirror images of the protein in the periodic cell. Can the authors provide an explanation for the choice of such a small number of lipids?

-          The membrane simulation was presumably performed at 300K (is this correct?). Why was this temperature chosen? Additionally, the melting point of PEA is higher than 300K. Does 300K enable the modelling of a fluid lipid environment around the channel for this lipid?

-          The section describing the PIIIA-Nav1.4 complex simulations (lines 248-290) provides some interesting observations from the trajectories, but is largely descriptive. It would strengthen this discussion immensely to provide some quantitative measure of pore exposure/blockage induced by the various isomers. Perhaps something as simple as a SASA for certain selected residues near the pore opening could show this more quantitatively?

-          The manuscript could benefit from some minor editing. Although it is very well-written, there are some overly casual expressions that should be corrected. Eg. Line 228-230: “Visual inspection of the MD trajectory of these isomers revealed huge amounts of rotational and translational motion as the isomers constantly tried finding a structurally and energetically favorable state.”

Author Response

file attached

Reviewer 2 Report

The manuscript by George et al., discusses the various structural conformations of u-PIIIA resulting from different disulfide bond arrangements and how these various conformations affect conotoxin potency Nav1.4.  The paper is very well written and most issues and concerns have been addressed.  I only have very minor suggestions/comments:

In Table 1, it may be easier for the reader to see the disulfide connectivities in form of sequences rather than numbers, such as shown in figure 3A, B and C.  The authors do not have to show the entire sequence of the peptides, but just the C-C connectivities.

On page 7, line 219, the sentence that starts with "The single linkage RMSD clustering...." is a bit confusing.  Can authors reword or revise the sentence to better clarify its meaning?

On page 12, line 336, space is needed between "heavily between".

In the Supplemental section, I don't think it's necessary to show both the molecular modeling/MD simulations for each toxin alone and with respect to native peptide number 2.  Just show the data comparing the analogs to native peptide 2.

Author Response

file attached

Reviewer 3 Report

Effect of conformational diversity on the bioactivity 3 of μ-conotoxin PIIIA disulfide isomers

by Paul George et al.

This paper reports the role played by conformational structure of µ -PIIIA conotoxin on the interaction with sodium currents. 

The paper is interesting and deals about a topic subject especially for what concern the potential use of these toxins in biomedicine. The authors have a solid biochemical background on this topic and the results are based on three different experimental approaches. So, nonetheless possible artifacts that authors discuss, I believe that the study adds a new contribute to the knowledge of the biological activity of these important compounds.

I found some minor aspects to be revised, or some parts that are a bit unclear, as listed below:
-authors do not well introduce the current knowledge on the conotoxin impact  on the ion current and especially on Nav as a general context. An effort should be made to better introduce the background of their experimental work by referring to the recent review by Tosti et al., 2017 on Marine Drugs which provides a deep overview on the μ -conotoxins and sodium current activity.

- lines 60 to 65 there is too much description of the results, which is not appropriate for the introduction section.  Move these sentences

- On the contrary in lines 77 to 81 authors describe the biology of µ -PIIIA which is not appropriate for the results section. Again move this description in the introduction.

- line 107 what is fig S2?

-line 130 typo (PIIIIA), correct it

In conclusion the paper is well written, the experiments are well designed and reveals consistent results and the references are updated.  I support the publication of this MS on Marine Drugs

Author Response

file attached
